# Nutritional and Functional Properties of Defatted Flour, Protein Concentrates, and Isolates of *Brachytrupes membranaceus* (Orthoptera: Gryllidae) (Drury: 1773) and *Macrotermes subhyalinus* (Isoptera: Blattodea) (Rambur: 1842) from Burkina Faso

**DOI:** 10.3390/insects13090764

**Published:** 2022-08-24

**Authors:** Aminata Séré, Adjima Bougma, Bazoin Sylvain Raoul Bazié, Philippe Augustin Nikièma, Olivier Gnankiné, Imael Henri Nestor Bassolé

**Affiliations:** 1Département de Biochimie Microbiologie, Université Joseph KI-Zerbo, Ouagadougou 03 B.P. 7021, Burkina Faso; 2Département de Biologie et de Physiologie Animales, Université Joseph KI-Zerbo, Ouagadougou 03 B.P. 7021, Burkina Faso

**Keywords:** *Brachytrupes membranaceus*, Macrotermes subhyalinus, nutritional values, functional properties

## Abstract

**Simple Summary:**

Edible insects are a source of nutrients for local populations. The present study evaluated proximal composition, fatty acid, and mineral profiles of *Brachytrupes membranaceus* and *Macrotermes subhyalinus* from Burkina Faso. The digestibility and functional properties of their defatted flours, protein concentrates, and isolates were also determined. *Brachytrupes membranaceus* protein concentrates and isolates showed the best nutritional values. They were more digestible and endowed with functional properties. The results revealed that defatted flours, concentrates, and isolates of proteins of *Brachytrupes membranaceus* and *Macrotermes subhyalinus* were alternative sources of minerals, proteins, essential amino acids, and essential fatty acids. They could, therefore, be used to combat protein, iron, and zinc deficiencies and for the bio-fortification of foods.

**Abstract:**

*Brachytrupes membranaceus* and *Macrotermes subhyalinus* are edible insects in Burkina Faso. Our research aimed to evaluate the nutritional composition and functional properties of the defatted flours, protein concentrates, and isolates of *Brachytrupes membranaceus* and *Macrotermes subhyalinus*. Proximate and mineral composition were determined according to AOAC methods. The amino acid and fatty acid composition were determined by high-performance liquid chromatography (HPLC) and gas chromatography, respectively. The protein concentrates and isolates were obtained by solubilization, precipitation, and lyophilization. *Macrotermes subhyalinus* showed the highest protein (45.75 g/100 g), iron (11.76 mg/100 g), and zinc (13.18 mg/100 g) contents. The highest isoleucine and lysine contents, the best fat absorption (10.87 g/g), and foaming capacities (49.60%) were obtained with the isolate of *Brachytrupes membranaceus*. Consumption of *Macrotermes subhyalinus* could be used to fight or correct iron and zinc deficiencies. *Macrotermes subhyalinus* was a source of macronutrients and micronutrients, while the protein concentrates and isolates of *Brachytrupes membranaceus* were endowed with functional properties (fat absorption and foaming capacities).

## 1. Introduction

Protein concentrates and isolates are widely used in the food industry. Protein isolates are the most refined form of protein products, containing the highest concentration of protein. Unlike protein powders and concentrates, they do not contain dietary fiber [1]. Protein isolates are important sources of high lysine protein. They are very digestible, endowed with functional properties, and are ideal ingredients for the formulation of food products [1,2]. The main sources for protein concentrate and isolate extractions are cereals (wheat and rice), legumes, tubers, oilseeds (peanuts, soybeans, sesame and beans), nuts, milk, meat, and fish [2,3,4]. Protein concentrates and isolates are mainly extracted by precipitation, alkaline extraction, and ultrafiltration [1]. Isoelectric point precipitation is the most widely used method and consists of solubilizing the proteins in saline solution and/or in an environment where the pH is close to the isoelectric point, which favors the precipitation of the proteins of interest [5]. Alkaline extraction uses alkaline reagents and is primarily used for protein extraction from legumes. However, several changes, such as lysine destruction, lysine-alanine formation, and racemization, can occur during alkaline extraction and reduce protein quality [1,6]. Soy isolates are used to fortify all types of pasta, such as macaroni and spaghetti [7]. Legume isolates are widely used in the meat, snack, and dessert industries [8,9]. Functional properties of protein products are physicochemical indicators that determine the behavior of proteins in the production of food products. These properties are mainly linked to the structure and composition of amino acids of native proteins. However, physical and chemical treatments can change the functionality of proteins [10]. Protein products’ most important functional properties are water absorption capacity, oil absorption capacity, foaming capacity, emulsion stability, foam stability, and gel-forming capacity [11,12]. The functional properties of milk, eggs, and soy have been extensively studied [13,14]. However, little is known about the functional properties of invertebrates, such as edible insects.

Insects that belong to the Orthoptera order include more than 278 members, such as locusts, crickets, and grasshoppers [15]. Members of the Orthoptera order are rich in lipids (2.49–53.05%), proteins (12.1–77.13%), and minerals (0.34–9.36%) [16]. Grasshoppers contain 50.50% protein, 15.30% crude fiber, and 6.40% ash. They are also rich in calcium (146%), magnesium (56.40%), iron (0.06 mg/kg) and zinc (0.04 mg/kg) [17]. *Ruspolia differens* (brown), *Ruspolia differens* (green), and *Homorocoryphus nitidulus* have linoleic acid contents between 29.50 and 45.63%. *Chondracis roseapbrunner* contains up to 40.10% linolenic acid. Members of the *Orthoptera* order are sources of zinc (12.00–78.00 mg/100 g) and iron (0.35–1562 mg/100 g).

Insects that belong to the *Isoptera* order include more than 59 members [15]. Members of the *Isoptera* order are rich in proteins (20.40–65.62%), lipids (21.35–46.10%), and linoleic acid (1.90–11.26%), with very high iron contents [16].

*Brachytrupes membranaceus* and *Macrotermes subhyalinus* are two common species of edible insects in Burkina Faso [18]. *Brachytrupes membranaceus* is mainly consumed in the Western part of Burkina Faso, while *Macrotermes subhyalinus* is consumed everywhere in the country. Both species are consumed by the Bobo, Guin, Mossi, Fulani, Sambla, Senoufo, Toussian, and Turka ethnic groups. *Brachytrupes membranaceus* is mainly available from September to October, while *Macrotermes subhyalinus* is available from June–July [18]. These two species are also eaten in Nigeria, Angola, Togo, and Zambia [19]. *Brachytrupes membranaceus* and *Macrotermes subhyalinus* contain 35.06% and 38.42% protein, 53.05 and 46.10% fat, and 3.25 and 6.56% ash, respectively [16,20]. However, to the best of our knowledge, published data on the amino acid composition, digestibility, and functional properties of the defatting flour, protein concentrates, and isolates of *Brachytrupes membranaceus* and *Macrotermes subhyalinus* are scarce. These properties could be helpful for better use of insect products in the food industry [21]. The present study aimed to assess the nutritional potential, digestibility, and functional properties of defatting flour, protein concentrates, and isolates from *Brachytrupes membranaceus* and *Macrotermes subhyalinus*.

## 2. Materials and Methods

### 2.1. Sample Collection and Pretreatment of Insects

Adults of *Brachytrupes membranaceus* and *Macrotermes subhyalinus* were collected in Dindéresso (11°13′60″ N; 4°25′60″ W) and Koro (11°08′40″ N; 4°11′55″ W) in September and August 2018, respectively. They were identified at the Environment and Forests Department (DEF) of the National Center for Scientific and Technological Research (CNRST), using the method described by Scholtz [12]. Two kilograms (2 kg) of each species were collected in their natural habitat. The collected insects were immediately placed in coolers containing ice and transported to the laboratory. The samples were washed thoroughly in distilled water, oven-dried (50 °C), and ground to a powder for further analysis.

### 2.2. Preparation of Protein Concentrate and Isolates

Wolf’s method with minor modifications was used to extract protein concentrates and isolates [22]. Before the extraction, the flour of the insect was defatted and stirred for 1 to 2 h at room temperature. Protein concentrates and isolates were extracted by centrifugation (10,000× *g* for 30 min at 4 °C) from the supernatant and pellet in acidic (pH 4.5) and alkaline (pH 11) solutions, respectively. Both protein concentrates and isolates were then washed with de-ionized water, re-dissolved in de-ionized water, neutralized to pH 7.0 with 1N NaOH at room temperature, and then freeze-dried.

### 2.3. Proximate and Mineral Compositions

Determination of moisture, fat, protein, and ash contents was carried out according to AOAC methods 950.46, 960.39, 979.09, and 920.153, respectively [23]. Energy value was obtained according to the method described by Merrill and Watt [24]. The Ca, Na, K, Mg, Zn, and Fe contents were determined according to the AOAC method 999.11 [23] using an atomic spectrophotometer (Varian AA240 FS, Varian Inc., Palo Alto, CA, USA). Before the determination of mineral concentrations, the samples were incinerated at 550 °C and acid digested.

### 2.4. Fatty Acid Composition

The fatty acid composition was determined following the method described by IUPAC [25]. Fatty acid methyl esters were prepared following the Khan method [26]. Fatty acids, in the form of their methyl esters, were analyzed on a capillary column (60 m ID: 0.25 mm, film: 0.25 µm, J&W Scientific Co., Folsom, CA, USA) by gas chromatography (Agilent Technologies 6890N, Agilent Technologies, Inc., Santa Clara, CA, USA). The identification of the representative peaks of the fatty acid methyl esters was carried out, using reference fatty acid methyl esters, by comparing the retention distances of each peak in the chromatogram with those obtained by the standards.

### 2.5. Amino Acid Composition of Defatting Flour, Protein Concentrates, and Isolates

The amino acid composition of defatted flour, protein concentrates, and isolates was determined by reverse-phase HPLC using the Pico Tag method, as described by Bidlingmeyer et al. [27]. The samples were defatted with n-hexane (Sigma Aldrich, Saint Louis, MO, USA) and hydrolyzed with 6N hydrochloric acid ((Carlo Erba, Val de Reuil, France), 37%), for 24 h at 110 °C, then filtered and derivatized with phenylisothiocyanate. The amino acid derivatives were separated by HPLC and detected by a UV detector at 254 nm after elution through a Pico Tag precolumn (Nova-Pak C18 Guard Column, 60Å, 4 μm, 3.9 mm × 20 mm, Waters Corp., Milford, MA, USA) and column (C18 PICO’TAG Column Waters (3.9 × 150 mm)), according to the conditions described by Bidlingmeyer et al. [27].

### 2.6. Protein Digestibility

The protein digestibility was assessed according to the methods described by Hsu et al. [28] and Satterlee et al. [29]. Ten milliliters of an aqueous protein suspension (1 mg per mL distilled water) were equilibrated at 37 °C to pH 8.0. One milliliter of three-enzyme solution (1.61 mg trypsin (Sigma aldric, Saint Louis, MO, USA), 3.96 mg chymotrypsin (MP Biomedicals LLC, Illkirch, France), and 2.36 mg peptidase (Megazyme, Bray, Ireland) per mL) was added to the protein suspension, and after exactly 10 min of incubation, the pH was recorded. The percent of protein digestibility (Y) was obtained from the following equation: digestibility = 74.33 + 53.21X

X is the volume of NaOH (ml) poured at T = 10 min to maintain the pH at 8.0.

### 2.7. Water Absorption Capacity

The sample’s water absorption capacity (WAC) was determined using the method outlined by Diniz and Martin [30]. Briefly, 0.5 g of dried sample was rehydrated with 20 mL of distilled water, stirred with a vortex mixer for 30 s, then centrifuged at 8000× *g* for 15 min. The difference between the final and initial weight of the protein sample was determined and the results were reported as g of water absorbed per g of protein sample.

### 2.8. Fat Absorption Capacity

The method of Haque and Mozaffar [31] was used to determine fat absorption capacity (FAC). Briefly, 0.5 g of dry sample was dispersed in 10 mL of vegetal oil, vortexed for 15 s, then centrifuged at 8000× *g* for 15 to decant the free oil. The fat absorption capacity was expressed as g of fat adsorbed per g of protein sample.

### 2.9. Foaming Capacity

Foaming capacity (FC) was determined according to the method of Guo et al. [32]. First, 20 mL of 1.0% protein sample solution was homogenized at 16,000 rpm for 2 min. FC was calculated from the following equation: $$\mathrm{FC}\;\left(\%\right)=\frac{\mathrm{VO}-V}{V}$$

V is the volume before whipping (mL); V0 is the volume after whipping (mL).

### 2.10. Statistical Analysis

Analytical determinations were performed in triplicate. The values of the different parameters have been expressed as mean and standard deviation (SD). Significant differences between the mean values (*p* < 0.05) were determined using ANOVA using XLSTAT software for Windows (XLSTAT 2016.02.27444).

## 3. Results

### 3.1. Proximate Composition of Brachytrupes membranaceus and Macrotermes subhyalinus

Proximate composition, expressed on a dry matter basis, of *Brachytrupes membranaceus* and *Macrotermes subhyalinus* is shown in Table 1. The protein content of *Brachytrupes membranaceus* (26.44%) was two-fold lower than that of *Macrotermes subhyalinus* (45.75%). Both species exhibited similar lipid (49.56–50.12%) contents. *Brachytrupes membranaceus* showed higher carbohydrate and ash contents, while *Macrotermes subhyalinus* had a higher energy value. The protein and fat contents of *Brachytrupes membranaceus* were lower than those previously reported by Agbidye et al. [33]. The ash, protein, and lipid contents of *Macrotermes subhyalinus* were higher and lower than those reported for *Macrotermes nigeriensis* [34] and cricket [35], respectively.

### 3.2. Mineral Composition of Brachytrupes membranaceus and Macrotermes subhyalinus

The mineral compositions of *Brachytrupes membranaceus* and *Macrotermes subhyalinus* are reported in Table 2. The highest levels of iron, zinc, potassium, and sodium were obtained with *Macrotermes subhyalinus.* The calcium and magnesium contents of *Brachytrupes membranaceus* were 2.5 and 1.5-fold higher than those of *Macrotermes subhyalinus*, respectively.

The zinc and calcium contents of *Brachytrupes membranaceus* were higher than those of *Brachytrupes orientalis*, while the iron, magnesium, potassium, and sodium contents were lower than those of the same species [36]. Shah and Wanapat [35] reported high levels of iron and zinc in crickets (11.6 and 21.5 mg/100 g, respectively). Akullo et al. [37] reported higher iron, potassium, sodium, and zinc levels in *Macrotermes bellicosus.* The iron and zinc contents of *Macrotermes subhyalinus* can cover the recommended daily intake for adults [38]. Both *Brachytrupes membranaceus* and *Macrotermes subhyalinus* could fight or correct iron and zinc deficiencies. The low sodium contents of *Brachytrupes membranaceus* and *Macrotermes subhyalinus* allow their use in low salt diets [16].

### 3.3. Cholesterol Content and Fatty Acid Profile of Brachytrupes membranaceus and Macrotermes subhyalinus

The cholesterol content was 0.98 and 1.47 g/100 g of oil for *Macrotermes subhyalinus* and *Brachytrupes membranaceus,* respectively (Table 3). The cholesterol content of *Macrotermes subhyalinus* was higher than that of *Macrotermes bellicosus* [39].

*Brachytrupes membranaceus* and *Macrotermes subhyalinus* contained 37.38 and 40.96% saturated fatty acids (SFA), 44.52 and 51.75% monounsaturated (MUFA), and 18.11 and 7.26% polyunsaturated fatty acids (PUFA), respectively (Table 3). Oleic (42.78–50.04%), palmitic (19.89–27.26%), stearic (14.79–11.93%), and linoleic (7.01–14.87%) acids were the main fatty acids found in the oils of both *Brachytrupes membranaceus* and *Macrotermes subhyalinus*. These four fatty acids made up over 90% of the total fatty acids. Palmitic and stearic acid contents of *Brachytrupes membranaceus* were lower than that of *Brachytrupes orientalis* [36]. The contents of stearic and oleic acids of *Macrotermes subhyalinus* were higher than that of *Macrotermes bellicosus* [39]. Both Brachytrupes membranaceus and Macrotermes subhyalinus are sources of linoleic acid (7.01–14.87%, respectively). Brachytrupes membranaceus also contained 3.11% linolenic acid. Linolenic and linoleic acids are strictly essential, as they are not synthesized by the body and must, therefore, be supplied in full by the diet [40]. In the body, linolenic acid is converted into eicosapentaenoic acid, then into docosahexaenoic acid, and linoleic acid into dihomo-gamma-linolenic acid, then into arachidonic acid [41]. These are subsequently transformed into prostaglandins and leukotrienes.

### 3.4. Proteins Contents of Protein Concentrates and Isolates

Protein contents of the protein isolate ranged from 88.68 to 89.32%, while the protein concentrates varied from 50.96 to 63.82% (Table 4). Both the protein concentrates and isolates of *Brachytrupes membranaceus* had 1.92 and 3.37-fold higher protein contents than those of the defatted flour. A similar increase in protein contents of the protein concentrates and isolates has been reported for *Schistocerca gregaria* and *Apis mellifera* isolates [42]. The difference in the protein content increase in the isolates and concentrates compared to the defatted flour of *Brachytrupes membranaceus* and *Macrotermes*
*subhyalinus* could be due to diverging extraction methods. Indeed, the alkaline extraction-isoelectric precipitation method improves the protein content [4]. The production of protein concentrates and isolates consists of aqueous solubilization of proteins and carbohydrates at neutral or alkaline pH and the selective recovery of the solubilized protein. Factors such as pH, presence (absence) of salts and their concentration, ionic strength of the medium, net charge, and electrostatic repulsions influence proteins’ yield and extraction properties [43].

### 3.5. Amino Acid Composition of Brachytrupes membranaceus and Macrotermes subhyalinus

The amino acid composition of *Brachytrupes membranaceus* and *Macrotermes subhyalinus* is shown in Table 5 and Table 6, respectively. Both species contained all essential amino acids. Lysine (9.18–13.91 g/100 g protein), isoleucine (8.31–9.54 g/100 g protein), phenylalanine + tyrosine (6.82–8.43 g/100 g protein) and threonine (5.50–6.23 g/100 g protein) were the most abundant essential amino acids in defatted flour, protein concentrates and isolates of *Brachytrupes membranaceus.* Threonine (9.90–10.47 g/100 g protein), leucine (8.43–9.44 g/100 g protein), and phenylalanine + tyrosine (6.22–8.92 g/100 g protein) were the predominant essential amino acids in defatted flour, protein concentrates and isolates of *Macrotermes subhyalinus.* Séré et al. [44] reported a similar increase in leucine, isoleucine, and lysine content in the protein isolates of *Carbula marginella* and *Cirina butyrospermi*.

Leucine and valine were limiting amino acids in defatted flour, protein concentrates, and isolates of *Brachytrupes membranaceus,* while lysine was a limiting amino acid in defatted flour of *Macrotermes subhyalinus* [45]. Essential amino acid contents of *Macrotermes subhyalinus* were higher than those of *Macrotermes bellicosus* and *Macrotermes nigierensis* [46]. Interestingly, protein concentrates and isolates from *Brachytrupes membranaceus* and *Macrotermes subhyalinus* had higher lysine levels than those recommended by FAO for the daily requirement of adults [45]. Lysine levels are generally low in most cereal proteins, which are staple foods in developing countries. It is also the limiting amino acid in most diets [47]. The high lysine values in the concentrates and isolates of *Brachytrupes membranaceus* and *Macrotermes subhyalinus* suggest that they can be used as dietary supplements.

### 3.6. Protein Digestibility

Protein digestibility ranged from 79.92% to 82.57%; from 82.31 to 83.37% and from 84.17 to 85.24% for the defatted flour, protein concentrates, and isolates, respectively. The protein isolates of both *Brachytrupes membranaceus* and *Macrotermes subhyalinus* exhibited the highest protein digestibility (Table 7). Oibiokpa et al. [46] reported similar digestibility for *Gryllus assimilis*, *Cirina forda*, *Melanoplus foedus* and *Macrotermes nigeriensis*. The high digestibility associated with the isolate fractions could be explained by the reduction in the proteolytic enzyme inhibitor during the extraction process.

### 3.7. Water Absorption Capacity of Brachytrupes membranaceus and Macrotermes subhyalinus

The water absorption capacity of *Brachytrupes membranaceus* and *Macrotermes subhyalinus* varied from 3.68% to 1.72% for defatted flour, from 4.68% to 4.11% for protein concentrates, and from 4.51% to 3.43% for protein isolates, respectively (Table 8 and Table 9). Both protein concentrates and isolates of *Brachytrupes membranaceus* had the highest water absorption capacity. The highest protein contents of both protein concentrates and isolates of *Brachytrupes membranaceus* could be due to the higher protein content of *Brachytrupes membranaceus* than *Macrotermes subhyalinus.* The water absorption capacity of defatted flour of *Brachytrupes membranaceus* was higher than that of the whole insect of *Acheta domesticus* (2.03 g/g) [48]. The water absorption capacity of the protein isolates of *Brachytrupes membranaceus* was higher than that obtained with the isolates of *Schistocerca gregaria* and *Gryllodes sigillatus* [21]. The highest water absorption capacities were obtained for protein concentrates of *Brachytrupes membranaceus* and *Macrotermes subhyalinus*. This could be because concentrates include carbohydrates that absorb water too [4]. The differences in water absorption obtained in the present study could be related also to the content of hydrophilic amino acids, the presence of non-protein components, and the type, quality, and conformation of the proteins [49,50].

### 3.8. Fat Absorption Capacity of Brachytrupes membranaceus and Macrotermes subhyalinus

The values of fat absorption capacity of the defatted flour, protein concentrates and isolates varied from 2.03% to 10.87% (Table 8 and Table 9). Protein isolates of *Brachytrupes membranaceus* had the highest fat absorption capacity (10.87%), while the defatting flour of *Macrotermes subhyalinus* had the lowest one (2.03%). The fat absorption capacity of the defatting flour of *Brachytrupes membranaceus* was higher than that of *Gryllodes sigillatus* (2.82 g/g) and *Schistocerca gregaria* (Zielińska et al. [21]), but similar to that of *Acheta domesticus* [48]. Torruco-Uco et al. [51] reported values lower than that of *Brachytrupes membranaceus* with *Sphenarium purpurascens* (2.79 g/g at 60–70 °C). The fat absorption capacity of the protein isolate of *Brachytrupes membranaceus* was higher than that obtained with the isolates of *Gryllodes sigillatus* and *Schistocerca gregaria* [21].

The low-fat absorption capacity of the isolate of *Macrotermes subhyalinus* could be due to its low hydrophobic amino acid content (25.77 mg/100 g protein) compared to the protein isolate of *Brachytrupes membranaceus,* which had a high hydrophobic amino acid content (33.59 mg/100 g protein). Although the protein isolate of *Brachytrupes membranaceus* had the highest fat absorption capacity, it had a low hydrophobic amino acid content (33.59 mg/100 g protein) compared to that of the concentrates of *Brachytrupes membranaceus* (40.34 mg/100 g protein). This could be explained by the location of hydrophobic amino acid residues on the protein surface of the protein isolate of *Brachytrupes membranaceus* [52]. The fat absorption capacity is the ability of proteins to physically bind to fat through capillary attraction. It is due to the presence of electrostatic interactions, hydrophobic forces, and hydrogen bonds, which are the forces involved in lipid–protein interactions [53]. Knowledge of oil absorption capacity is important in food technology, as it imparts certain characteristics to the product, such as flavor retention palatability enhancement, and an increase in shelf life by reducing humidity and fat loss [54].

### 3.9. Foaming Capacity of Brachytrupes membranaceus and Macrotermes subhyalinus

The foaming capacity ranged from 12.2% to 30%, from 11.8% to 39.4%, and from 3.6% to 49.6% for defatted flour, and protein isolates and concentrates, respectively (Table 8 and Table 9). The foaming capacity of the protein isolates of *Brachytrupes membranaceus* was 13.77-fold higher than that of protein isolates of *Macrotermes subhyalinus*. The foaming capacity of defatted flour of *Brachytrupes membranaceus* was higher than that reported by Zielińska et al. [21] with whole insects of *Schistocerca gregaria* (22.33%). The foaming capacity of the *Brachytrupes membranaceus* protein isolates was higher and lower than that of *Schistocerca gregaria* (32.00%) and protein *Gryllodes sigillatus* isolates (99.00%), respectively [21]. Although the defatted flour and the protein concentrates of *Macrotermes subhyalinus* had high protein contents compared to those of *Brachytrupes membranaceus*, these proteins were not endowed with functional properties. This could be explained by the fact that the protein concentrates and isolates contained higher levels of hydrophobic amino acids (40.34 and 33.59 mg/100 g protein, respectively). Foaming capacity depends on proteins and other components, such as carbohydrates, the location of amino acid residues on the surface of the protein, and surface hydrophobicity [55].

## 4. Conclusions

The present study determined the nutritional and functional properties of defatted flours, concentrates, and protein isolates of *Brachytrupes membranaceus* and *Macrotermes subhyalinus*. *Macrotermes subhyalinus* was a rich source of proteins, lipids, iron, and zinc. Defatted flours, protein concentrates, and protein isolates of *Brachytrupes membranaceus* and *Macrotermes subhyalinus* were sources of essential amino acids. Protein concentrates and isolates of *Brachytrupes membranaceus* have high fat absorption and foaming capacities. *Brachytrupes membranaceus* and *Macrotermes subhyalinus* can be recommended as nutritional and functional supplements. Tasting tests could be set up to assess the flavor and acceptability of protein concentrates and isolates of edible insects.

## Figures and Tables

**Table 1 insects-13-00764-t001:** Proximate composition on a dry basis (g/100 g) and energy (Kcal/100 g) of Brachytrupes membranaceus and Macrotermes subhyalinus.

Parameters	*Brachytrupes membranaceus*	*Macrotermes subhyalinus*
Moisture (wet basis)	43.20 ± 2.42 ^b^	50.66 ± 0.28 ^a^
Ash	3.75 ± 0.04 ^a^	3.20 ± 0.05 ^b^
Crude protein	26.44 ± 0.30 ^b^	45.75 ± 0.32 ^a^
Crude fat	49.56 ± 0.20 ^a^	50.12 ± 0.11 ^a^
Carbohydrates	20.23 ± 0.53 ^a^	0.92 ± 0.41 ^b^
Energy	632.82 ± 0.49 ^b^	637.81 ± 0.55 ^a^

^a,b^ Means in the same row with different superscripts are significantly different (*p* < 0.05).

**Table 2 insects-13-00764-t002:** Mineral composition (mg/100 g) of *Brachytrupes membranaceus* and *Macrotermes subhyalinus*.

Minerals	*Brachytrupes membranaceus*	*Macrotermes subhyalinus*
Calcium	193.45 ± 0.02 ^a^	74.62 ± 0.89 ^b^
Magnesium	75.39 ± 0.00 ^a^	49.86 ± 0.09 ^b^
Potassium	522.22 ± 0.01 ^b^	635.61 ± 0.25 ^a^
Sodium	61.69 ± 0.01 ^a^	74.82 ± 0.97 ^b^
Iron	7.84 ± 0.01 ^b^	11.76 ± 0.19 ^a^
Zinc	9.95 ± 0.01 ^b^	13.18 ± 0.09 ^a^

^a,b^ Means in the same row with different superscripts are significantly different (*p* < 0.05).

**Table 3 insects-13-00764-t003:** Cholesterol content (g/100 g fat) and fatty acid composition (%) of *Brachytrupes membranaceus* and *Macrotermes subhyalinus*.

Cholesterol/Fatty Acids	*Brachytrupes membranaceus*	*Macrotermes subhyalinus*
Cholesterol	1.47 ± 0.006 ^a^	0.98 ± 0.006 ^b^
Caproic acid	0.11 ± 0.00 ^a^	0.08 ± 0.04 ^a^
Capric acid	0.18 ± 0.00 ^b^	0.09 ± 0.00 ^b^
Lauric acid	0.11 ± 0.01 ^b^	0.07 ± 0.01 ^b^
Myristic acid	0.71 ± 0.00 ^a^	0.38 ± 0.00 ^b^
Myristoleic acid	0.00 ± 0.00 ^b^	0.50 ± 0.01 ^a^
Palmitic acid	19.89 ± 0.01 ^b^	27.26 ± 0.02 ^a^
Palmitoleic acid	0.93 ± 0.02 ^b^	1.19 ± 0.01 ^a^
Margaric acid	0.60 ± 0.00 ^a^	0.37 ± 0.01 ^b^
Stearic acid	14.79 ± 0.01 ^a^	11.93 ± 0.03 ^b^
Oleic acid	42.78 ± 0.13 ^b^	50.04 ± 0.10 ^a^
Linolelaidic acid	0.00 ± 0.00 ^b^	0.05 ± 0.00 ^a^
Linoleic acid	14.87 ± 0.11 ^a^	7.01 ± 0.03 ^b^
Linolenic acid	3.11 ± 0.01 ^a^	0.08 ± 0.02 ^b^
Arachidic acid	0.59 ± 0.00 ^a^	0.51 ± 0.01 ^b^
Gondoic acid	0.80 ± 0.02 ^a^	0.00 ± 0.00 ^b^
Docosapentaenoic acid	0.13 ± 0.01 ^a^	0.11 ± 0.05 ^a^
Lignoceric acid	0.38 ± 0.01 ^a^	0.25 ± 0.07 ^a^
TOTAL	100	99.94
SFA	37.38	40.96
MUFA	44.52	51.75
PUFA	18.11	7.26
SFA/MUFA	0.84	0.79

^a,b^ Means in the same row with different superscripts are significantly different (*p* < 0.05). SFA: saturated fatty acid, MUFA: monounsaturated fatty acid, PUFA: polyunsaturated fatty acid.

**Table 4 insects-13-00764-t004:** Protein contents (%) of defatted flour, concentrates and isolates *of Brachytrupes membranaceus* and *Macrotermes subhyalinus*.

Species	Proteins (%)		
	Defatted Flour	Protein Concentrate	Protein Isolate
** *B. membranaceus* **	26.44 ± 0.30 ^c^	50.96 ± 0.51 ^b^	89.32 ± 0.85 ^a^
** *M. subhyalinus* **	45.75 ± 0.32 ^c^	63.82 ± 0.64 ^b^	88.68 ± 0.68 ^a^

^a,b,c^ Means in the same row with different superscripts are significantly different (*p* < 0.05).

**Table 5 insects-13-00764-t005:** Amino acid composition (g/100 g protein) of defatted flour, protein concentrate, and isolate of Brachytrupes membranaceus.

Proteins/Amino Acids	Defatted Flour	Protein Concentrate	Protein Isolate
Histidine	1.82 ± 0.04 ^b^	1.53 ± 0.06 ^c^	2.22 ± 0.11 ^a^
Threonine	5.88 ± 0.09 ^b^	6.23 ± 0.27 ^a^	5.50 ± 0.31 ^b^
Valine	1.03 ± 0.03 ^b^	1.28 ± 0.10 ^a^	1.00 ± 0.03 ^b^
Methionine + cysteine	3.68 ± 0.10 ^c^	4.50 ± 0.37 ^b^	4.77 ± 0.17 ^a^
Isoleucine	8.35 ± 0.10 ^b^	8.31 ±0.21 ^b^	9.54 ± 0.49 ^a^
Leucine	2.72 ± 0.06 ^c^	2.89 ± 0.12 ^b^	4.10 ± 0.00 ^a^
Lysine	9.18 ± 0.06 ^c^	10.84 ± 0.80 ^b^	13.91 ± 0.67 ^a^
Phenylalanine + tyrosine	7.77 ± 0.08 ^b^	8.43 ± 0.41 ^a^	6.82 ± 0.49 ^c^
Tryptophane	ND	ND	ND
Aspartic acid and asparagine	5.71 ± 0.61 ^c^	7.80 ± 0.00 ^b^	13.08 ± 0.00 ^a^
Glutamic acid and glutamine	9.38 ± 0.19 ^c^	10.80 ± 0.12 ^b^	11.60 ± 0.32 ^a^
Serine	5.33 ± 0.01 ^a^	4.81 ± 0.18 ^c^	5.04 ± 0.39 ^b^
Glycine	10.97 ± 0.07 ^b^	11.86 ± 0.49 ^a^	9.76 ± 0.82 ^c^
Alanine	14.27 ± 0.05 ^b^	14.93 ± 0.66 ^a^	7.36 ± 1.02 ^c^
Arginine	12.93 ± 0.02 ^a^	3.95 ± 0.10 ^b^	3.77 ± 0.28 ^c^
Proline	1.00 ± 0.03 ^c^	1.85 ± 0.01 ^b^	1.54 ± 0.07 ^a^
Essential amino acids	40.43	44.00	47.86
Non-essential amino acids	59.57	56.00	52.14

^a,b,c^ Means in the same row with different superscripts are significantly different (*p* < 0.05).

**Table 6 insects-13-00764-t006:** Amino acid composition (g/100 g protein) of defatted flour, protein concentrate, and isolate of *Macrotermes subhyalinus*.

Amino Acids	Defatted Flour	Protein Concentrate	Protein Isolate
Histidine	2.00 ± 0.01 ^b^	2.24 ± 0.32 ^a^	1.68 ± 0.01 ^c^
Threonine	10.47 ± 0.02 ^b^	13.76 ± 0.00 ^a^	9.90 ± 0.02 ^c^
Valine	6.05 ± 0.00 ^a^	0.20 ± 0.03 ^c^	5.08 ± 0.03 ^b^
Methionine + cysteine	2.56 ± 0.01 ^a^	1.22 ± 0.19 ^b^	2.44 ± 0.05 ^a^
Isoleucine	1.58 ± 0.00 ^c^	3.89 ± 0.61 ^a^	2.83 ± 0.02 ^b^
Leucine	8.43 ± 0.00 ^c^	9.44 ± 1.40 ^a^	8.64 ± 0.04 ^b^
Lysine	1.71 ± 0.01 ^c^	10.66 ± 1.69 ^b^	12.03 ± 0.04 ^a^
Phenylalanine + tyrosine	6.22 ± 0.00 ^b^	8.92 ± 0.56 ^a^	6.37 ± 0.23 ^b^
Tryptophane	ND	ND	ND
Aspartic acid and asparagine	14.74 ± 0.00 ^a^	9.88 ± 0.00 ^c^	11.69 ± 0.79 ^b^
Glutamic acid and glutamine	12.73 ± 1.48 ^a^	10.24 ± 0.00 ^c^	11.18 ± 0.40 ^b^
Serine	7.38 ± 0.02 ^a^	7.07 ± 1.09 ^b^	4.78 ± 0.04 ^c^
Glycine	12.95 ± 0.03 ^a^	9.11 ± 1.44 ^c^	12.35 ± 0.01 ^b^
Alanine	5.21 ± 0.02 ^a^	4.62 ± 0.75 ^b^	3.71 ± 0.03 ^c^
Arginine	6.24 ± 0.05 ^a^	0.74 ± 0.13 ^c^	2.10 ± 0.09 ^b^
Proline	1.73 ± 0.05 ^c^	8.01 ± 1.24 ^a^	5.22 ± 0.05 ^b^
Essential amino acids	39.02	50.33	48.97
Non-essential amino acids	60.98	49.67	51.03

^a,b,c^ Means in the same row with different superscripts are significantly different (*p* < 0.05). ND: Not detected.

**Table 7 insects-13-00764-t007:** Digestibility (%) of defatted flour, concentrates, and isolates of *Brachytrupes membranaceus* and *Macrotermes subhyalinus*.

Species	Digestibility (%)		
	Defatted Flour	Protein Concentrate	Protein Isolate
** *B. membranaceus* **	79.92 ± 0.75 ^c^	83.37 ± 0.75 ^b^	85.24 ± 0.37 ^a^
** *M. subhyalinus* **	82.57 ± 1.12 ^c^	82.31 ± 0.75 ^b^	84.17 ± 1.12 ^a^

^a,b,c^ Means in the same row with different superscripts are significantly different (*p* < 0.05).

**Table 8 insects-13-00764-t008:** Functional properties of defatted flour, protein concentrate, and protein isolate of *Brachytrupes membranaceus*.

Functional Properties	Defatted Flour	Protein Concentrate	Protein Isolate
**Water absorption capacity (g** **/** **g)**	3.64 ± 0.04 ^b^	4.68 ± 0.09 a	4.51 ± 0.00 ^a^
**Fat absorption capacity (g** **/** **g)**	3.17 ± 0.54 ^b^	3.17 ± 0.60 ^b^	10.87 ± 0.23 ^a^
**Foaming capacity (%)**	30.00 ± 0.00 ^c^	39.40 ± 0.84 ^b^	49.60 ± 0.56 ^a^

^a,b,c^ Means in the same row with different superscripts are significantly different (*p* < 0.05).

**Table 9 insects-13-00764-t009:** Functional properties of defatted flour, protein concentrate, and protein isolate of *Macrotermes subhyalinus*.

Functional Properties	Defatted Flour	Protein Concentrate	Protein Isolate
**Water absorption capacity (g** **/** **g)**	1.72 ± 0.05 ^c^	4.11 ± 0.07 ^a^	3.43 ± 0.83 ^b^
**Fat absorption capacity (g** **/** **g)**	2.03 ± 0.33 ^c^	3.41 ± 0.24 ^a^	3.03 ± 0.27 ^b^
**Foaming capacity (%)**	12.20 ± 0.28 ^a^	11.80 ± 0.28 ^b^	3.60 ± 0.56 ^a^

^a,b,c^ Means in the same row with different superscripts are significantly different (*p* < 0.05).

## Data Availability

Data are contained within the article.

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
