# Peer review of "Nutritional and Functional Properties of Defatted Flour, Protein Concentrates, and Isolates of Brachytrupes membranaceus (Orthoptera: Gryllidae) (Drury: 1773) and Macrotermes subhyalinus (Isoptera: Blattodea) (Rambur: 1842) from Burkina Faso"

_insects, 2022, doi:10.3390/insects13090764_

Round 1

Reviewer 1 Report

Dear authors 

The study is intresting but there is same suggestion 

1. use the update reference in the introduction and discuss section 

For example : 1.https://www.researchgate.net/publication/357571974_Nutritional_composition_of_various_insects_and_potential_uses_as_alternative_protein_sources_in_animal_diets_-A_review?_sg%5B0%5D=Oot2zPDB6_5ZAI_lNh6SFNXIykXrXo-yV7k_tMi9WP64mgqU-YMYe6R2giQkhrr7AuJQNss5oHBnMRauvGBjYQKMt5zsoN4YgV2yQ5nq.Rdrtqy7uX7Sh67_AyLPunZ7y7XYrY7C8kt6ynIiN1gwvcrWKQM5830uPo2d4Ls8AvbCDwtO4ELxzSk_nsgaVWw

2.https://www.researchgate.net/publication/356381109_Gryllus_testaceus_walker_crickets_farming_management_chemical_composition_nutritive_profile_and_their_effect_on_animal_digestibility?_sg%5B0%5D=Oot2zPDB6_5ZAI_lNh6SFNXIykXrXo-yV7k_tMi9WP64mgqU-YMYe6R2giQkhrr7AuJQNss5oHBnMRauvGBjYQKMt5zsoN4YgV2yQ5nq.Rdrtqy7uX7Sh67_AyLPunZ7y7XYrY7C8kt6ynIiN1gwvcrWKQM5830uPo2d4Ls8AvbCDwtO4ELxzSk_nsgaVWw

2. improve the English language 

3. improve the introduction and result section 

Author Response

Review 1

  1. use the update reference in the introduction and discuss section 

For example : 1.https://www.researchgate.net/publication/357571974_Nutritional_composition_of_various_insects_and_potential_uses_as_alternative_protein_sources_in_animal_diets_-A_review?_

2.https://www.researchgate.net/publication/356381109_Gryllus_testaceus_walker_crickets_farming_management_chemical_composition_nutritive_profile_and_their_effect_on_animal_digestibility?

The updated reference has been used in the introduction and discussion section 

  1. improve the English language 

The English language has been corrected throughout the manuscript

  1. improve the introduction and result section 

The introduction has been improved by adding new data. The results section has been improved by converting the figures into tables. Parts of the results have been re-writing

Reviewer 2 Report

Line 156-158: Please correct the sentence „The protein content of Brachytrupes membranaceus (26.44 %) was twofold higher than that of Macrotermes subhyalinus (45.75 %).

„higher“ should be changed in „lower“

Line 159-160: Please correct the sentence „Both species exhibited similar ash (3.20 - 3.75 %) and lipid (49.56 - 50.12%) contents. Brachytrupes membranaceus showed the highest carbohydrate content while Macrotermes subhyalinus had the highest energy value.“

In both cases „the highest“ should be changed in „higher“

Regardless of the statistically different energy values, they are in fact, similar.

From section 3.4. Proteins contents should be omitted "protein content in defatted flours" from graphics and discussion because it is expected in all flour and protein concentrate and isolate types.

In section “3.6. Water Absorption Capacity of Brachytrupes membranaceus and Macrotermes subhyalinus“ should be explain why the values of Water Absorption Capacity of isolates are lower than for concentrates.

Section "3.7. Fat Absorption Capacity of Brachytrupes membranaceus and Macrotermes subhyalinus" is confusingly written and discussed.

This part should be rewritten.

Section „4. Conclusions“ should emphasize the need for further investigation to better understand differences in results.

In my opinion, authors have to make corrections in all comparisons throughout the manuscript regarding the terms like "the highest" and "higher" and  "the lowerest" and "lower".

Please check grammar also.

Author Response

Review 2

  1. Line 156-158: Please correct the sentence „The protein content of Brachytrupes membranaceus(26.44 %) was twofold higher than that of Macrotermes subhyalinus (45.75 %).   „higher“ should be changed in „lower“

Line 156-158: The protein content of Brachytrupes membranaceus (26.44 %) was twofold lower than that of Macrotermes subhyalinus (45.75 %).

  1. Line 159-160: Please correct the sentence „Both species exhibited similar ash (3.20 - 3.75 %) and lipid (49.56 - 50.12%) contents. Brachytrupes membranaceusshowed the highest carbohydrate content while Macrotermes subhyalinus had the highest energy value.“In both cases „the highest“ should be changed in „higher“

Line 159-160: Both species exhibited similar ash (3.20 - 3.75 %) and lipid (49.56 - 50.12%) contents. Brachytrupes membranaceus showed the higher carbohydrate content while Macrotermes subhyalinus had

  1. Regardless of the statistically different energy values, they are in fact, similar.

The difference was statistically significative

  1. From section 3.4. Proteins contents should be omitted "protein content in defatted flours" from graphics and discussion because it is expected in all flour and protein concentrate and isolate types.

A correction has been made

  1. In section “3.6. Water Absorption Capacity of Brachytrupes membranaceusand Macrotermes subhyalinus“ should be explain why the values of Water Absorption Capacity of isolates are lower than for concentrates.

The explanation has been provided.

  1. Section "3.7. Fat Absorption Capacity of Brachytrupes membranaceus and Macrotermes subhyalinus" is confusingly written and discussed. This part should be rewritten.

This part has been rewritten

  1. Section „4. Conclusions“ should emphasize the need for further investigation to better understand differences in results.
  2. In my opinion, authors have to make corrections in all comparisons throughout the manuscript regarding the terms like "the highest" and "higher" and  "the lowerest" and "lower".

All comparisons throughout the manuscript regarding the terms like "the highest" and "higher" and  "the lowerest" and "lower" has been corrected

Reviewer 3 Report

This current manuscript investigated nutritional and functional properties of two kind of edible insects in Burkina Faso. Overall, a very detailed paper with excellent figures representing the significant results. The article is very well written and is of interest both to local industry and consumers due to the trend and need of more functional ingredient to promote health benefits and quality of  food products. I have some suggestions and questions below, but over all this is a useful and well-written manuscript, just some few minor points need to be addressed.

1. line 94, The Wolf method Should be The Wolf's Method

2.  line 106,  Please put information about the country of origin of  atomic spectrophotometer

3. line 128, inapproate line break

4. line 260 'de protein' not formal 

5. Line 354, ‘Defatted flours and protein and protein isolates’ should be ‘Defatted flours, protein concentrate and protein isolates’

6.  Line 351-360,  The conclusion is not insightful, Please provide more  suggestions? How about the flavour, it will be better to mention in this artical.

7. Corrections to minor methodological errors and text editing

Author Response

Review 3

  1. line 94, The Wolf method Should be The Wolf's Method

The Wolf method has been corrected in The Wolf's Method

  1. line 106,  Please put information about the country of origin of  atomic spectrophotometer

the country of origin of  atomic spectrophotometer has been added « USA »

  1. line 128, inapproate line break

Line 128 has been deleted

  1. line 260 'de protein' not formal

line 260 has been corrected

  1. Line 354, ‘Defatted flours and protein and protein isolates’ should be ‘Defatted flours, protein concentrate, and protein isolates’

Line 354 : Defatted flours, protein concentrate, and protein isolate

  1. Line 351-360, The conclusion is not insightful, Please provide more  suggestions? How about the flavour, it will be better to mention in this artical.

The conclusion has been improved

  1. Corrections to minor methodological errors and text editing

Corrections have been made throughout the manuscript

Round 2

Reviewer 1 Report

it's ok. accept

Author Response

Response to Reviewers

Review 1

Thanks you very much for your contribution.

Reviewer 2 Report

Line 167: „Both species exhibited similar ash (3.20 - 3.75 %)“, but values in Table 1 show that the difference is statistically significant.

Lines 230-232: „The difference in the protein content increase of isolates and concentrates compared to the defatted flour of Brachytrupes membranaceus and Macrotermes subhyalinus could be explained by the extraction methods used [4].“ This is a wrong interpretation of the original text from reference No. 4.

In section 3.6. Water Absorption Capacity of Brachytrupes membranaceus and Macrotermes subhyalinus, authors stated that explanation, why the values of Water Absorption Capacity of isolates are lower than for concentrates, has been provided. However, I could not find an explanation, just a comparison of obtained values.

Author Response

Review 2

  1. Line 167: „Both species exhibited similar ash (3.20 - 3.75 %)“, but values in Table 1 show that the difference is statistically significant.

Line 167 has been corrected: Both species exhibited similar lipid (49.56 - 50.12 %) contents. Brachytrupes membranaceus showed the highest carbohydrate and ash contents while Macrotermes subhyalinus had the highest energy value.”

  1. Lines 230-232: The difference in the protein content increase of isolates and concentrates compared to the defatted flour of Brachytrupes membranaceus and Macrotermes subhyalinus could be explained by the extraction methods used [4].“ This is a wrong interpretation of the original text from reference No. 4.

The explanation has been provided. “The production of protein concentrates and isolates consists of aqueous solubilization of proteins and carbohydrates at neutral or alkaline pH and the selective recovery of the solubilized protein. Factors such as pH, presence (absence) of salts and their concentration, ionic strength of the medium, net charge, and electrostatic repulsions influence the yield and extraction properties of proteins.”

  1. In section 3.6. Water Absorption Capacity of Brachytrupes membranaceus and Macrotermes subhyalinus, authors stated that explanation, why the values of Water Absorption Capacity of isolates are lower than for concentrates, has been provided. However, I could not find an explanation, just a comparison of obtained values.

The explanation has been provided.

“The differences in water absorption obtained in the present study could be related to the content of hydrophilic amino acids, the presence of non-protein components, and the type, quality, and conformation of the proteins”.

Round 3

Reviewer 2 Report

Line 167: „Brachytrupes membranaceus showed the highest carbohydrate and ash contents while Macrotermes subhyalinus had the highest energy value.”

The highest has to be changed in higher in both cases.

In section 3.6.

The sentence:

„The highest water absorption capacities were obtained for protein concentrates of Brachytrupes membranaceus and Macrotermes subhyalinus. Both protein concentrates and isolates of Brachytrupes membranaceus had the highest water absorption capacity.“

should be excluded from the text.

Include in the discussion sentences no. 1 and 2.

1.            The higher water absorption capacities were obtained for protein concentrates than for protein isolates of Brachytrupes membranaceus and Macrotermes subhyalinus.

It is well-known fact that concentrates include carbohydrates which absorb water too. Please, include this fact in this part of the discussion.

2.            Both protein concentrates and isolates of Brachytrupes membranaceus had a higher water absorption capacity in comparison to Macrotermes subhyalinus.

At first, the content of protein in Brachytrupes membranaceus is much higher than in Macrotermes subhyalinus. You should mention that to explain above-mentioned discussion.

Author Response

Insects 1821834

Dear Editor,

The manuscript has been highly improved thanks to the valuable comments of the referees. All the remarks and suggestions of the referees have been taken into account according to our comprehension.

Response to Reviewers

Reviewer 2

  1. Line 167: „Brachytrupes membranaceus showed the highest carbohydrate and ash contents while Macrotermes subhyalinus had the highest energy value.”

The highest has to be changed in higher in both cases.

Line 167: has been corrected“ Brachytrupes membranaceus showed higher carbohydrate and ash contents while Macrotermes subhyalinus had higher energy value.”

  1. In section 3.6.

The sentence :„The highest water absorption capacities were obtained for protein concentrates of Brachytrupes membranaceus and Macrotermes subhyalinus. Both protein concentrates and isolates of Brachytrupes membranaceus had the highest water absorption capacity.“

should be excluded from the text.

Include in the discussion sentences no. 1 and 2.

   The higher water absorption capacities were obtained for protein concentrates than for protein isolates of Brachytrupes membranaceus and Macrotermes subhyalinus.

It is well-known fact that concentrates include carbohydrates which absorb water too. Please, include this fact in this part of the discussion.

 Both protein concentrates and isolates of Brachytrupes membranaceus had a higher water absorption capacity in comparison to Macrotermes subhyalinus.

At first, the content of protein in Brachytrupes membranaceus is much higher than in Macrotermes subhyalinus. You should mention that to explain above-mentioned discussion.

Section 3.6. has been rewritted

3.6. Water Absorption Capacity of Brachytrupes membranaceus and Macrotermes subhyalinus

The water absorption capacity of Brachytrupes membranaceus and Macrotermes subhyalinus varied respectively from 3.68 % to 1.72 % for defatted flour, from 4.68 % to 4.11 % for protein concentrates, and from 4.51 % to 3.43 % for protein isolates (Table 8 and 9). Both protein concentrates and isolates of Brachytrupes membranaceus had the highest water absorption capacity. The highest protein contents of both protein concentrates and isolates of Brachytrupes membranaceus could be due to the higher protein content of Brachytrupes membranaceus than Macrotermes subhyalinus. The water absorption capacity of defatted flour of Brachytrupes membranaceus was higher than that of the whole insect of Acheta domesticus (2.03 g / g) [48]. The water absorption capacity of protein isolates of Brachytrupes membranaceus was higher than that obtained with the isolates of Schistocerca gregaria and Gryllodes sigillatus [21]. The highest water absorption capacities were obtained for protein concentrates of Brachytrupes membranaceus and Macrotermes subhyalinus. This could be because concentrates include carbohydrates which absorb water too [4]. The differences in water absorption obtained in the present study could be related also to the content of hydrophilic amino acids, the presence of non-protein components, and the type, quality, and conformation of the proteins [49, 50].
